# Characteristics and Circumstances of Adolescents Obtaining Abortions in the United States

**DOI:** 10.3390/ijerph21040477

**Published:** 2024-04-13

**Authors:** Doris W. Chiu, Ava Braccia, Rachel K. Jones

**Affiliations:** Guttmacher Institute, New York, NY 10038, USA; abraccia@guttmacher.org (A.B.); rjones@guttmacher.org (R.K.J.)

**Keywords:** abortion, adolescents, sexual and reproductive health, access to health, U.S. state laws, teenage pregnancy, family planning, policy, barriers to care

## Abstract

The purpose of this study is to describe the sociodemographic and situational circumstances of adolescents obtaining abortion in the United States prior to the *Dobbs* decision. We use data from the Guttmacher Institute’s 2021–2022 Abortion Patient Survey, a cross-sectional survey of 6698 respondents; our analytic sample includes 633 adolescents (<20 years), 2152 young adults (20–24 years), and 3913 adults (25+ years). We conducted bivariate analyses to describe the characteristics and logistical and financial circumstances of adolescents obtaining abortions in comparison to respondents in the other age groups. The majority of adolescents identified as non-white (70%), and 23% identified as something other than heterosexual. We found that 26% of adolescents reported having no health insurance, and two-thirds of adolescent respondents reported that somebody had driven them to the facility. Adolescents differed from adults in their reasons for delays in accessing care; a majority of adolescents (57%) reported not knowing they were pregnant compared to 43% of adults, and nearly one in five adolescents did not know where to obtain the abortion compared to 11% of adults. Adolescents were more likely than adults to obtain a second-trimester abortion, which has increased costs. This study found that this population was more vulnerable than adults on several measures. Findings suggest that adolescents navigate unique barriers with regard to information and logistics to access abortion care.

## 1. Introduction

Access to abortion is an essential component of comprehensive adolescent sexual and reproductive health care [1]. When the Supreme Court of the United States (US) handed down the decision in *Dobbs v. Jackson Women’s Health Organization* (*Dobbs*) in June 2022, it eliminated the federal constitutional right to abortion. In response, a number of states have banned or imposed gestational limits and other medically unnecessary restrictions on abortion [2], making care out of reach for many people. But even prior to *Dobbs*, many people in the United States had to overcome barriers in order to access abortion; these obstacles were magnified for vulnerable populations, including people of color, those with low incomes, and adolescents [3]. This paper focuses on adolescents, as there has been less attention in recent abortion research to the unique experiences of this population.

In the year prior to *Dobbs*, adolescents, defined by the World Health Organization and in this paper as individuals under the age of 20 [4,5], accounted for approximately 10% of people obtaining abortions [6,7]. This means that in 2020, the last year for which there are comprehensive national statistics on abortion incidence [8], approximately 93,000 abortions were among adolescents.

Most of the recent research on US adolescent abortion patients is restricted to minors (under 18 years), usually in the context of understanding the impact of parental involvement laws [9,10,11]. There is scant research focusing on the larger population of adolescent abortion patients and their sociodemographic characteristics or experiences obtaining abortion. Much of what we know comes from studies where outcomes are provided by age group, which can vary from study to study, so broad findings are difficult to synthesize. We provide a brief summary of key studies examining the financial and logistical aspects of abortion care and identifying key differences among adolescents. The larger purpose of this paper is to describe the demographic characteristics and situational circumstances of adolescents obtaining abortions in the US and how they may be similar to, or different from, older abortion patients.

### Prior Research

Research has found that cost is the most common barrier to abortion care [12,13,14]. Public health insurance provided under Medicaid is the most common type of coverage for abortion patients [6,14]. However, the Hyde Amendment prohibits the use of federal Medicaid funds for abortion, and in most states, people with Medicaid coverage are unable to use it to pay for these health services [15]. In turn, most abortion patients pay out of pocket [6,16], often with great difficulty. On average, people reported paying $478 for abortion care in 2021–2022, and more than one-third of abortion patients also reported having to pay for transportation (37%) [17]. In order to cover this and related costs, nearly half (48%) of abortion patients reported having to adopt strategies such as delaying expenses and/or selling something [17].

These financial barriers are more likely to be pronounced for adolescents. Many young people with public and private health insurance obtain it through their parents, and concerns about confidentiality may prevent them from using insurance to pay for care even when it is allowed [18,19]. Adolescents, especially those still in school, may not be employed [20] or may work minimum-wage or part-time jobs [21]. These circumstances can make it difficult to come up with $500 to pay for an abortion—more if they have to cover travel or other costs—concerns that are especially relevant after the *Dobbs* decision. Currently, it is unknown to what extent adolescents can specifically adopt strategies such as delaying expenses and selling something in order to raise money in these circumstances.

The most common reason individuals report delays in abortion care is because they did not realize they were pregnant [6,22]. Some adolescents may not know or recognize the signs of pregnancy due to inadequate sex education or less exposure to pregnancy experiences [22]. Many adolescents have irregular periods [23], and a missed period may not be a cause for concern. These dynamics help explain why several studies have documented that adolescents experience longer delays in accessing care than adults [24,25].

Once an individual confirms they are pregnant and wants, or considers having, an abortion, they have to determine the availability of care, including where they can obtain an abortion, how they will travel to the facility, and how they will pay. In 2020, 38% of women aged 15–44 years lived in a county that did not have an abortion clinic [8]. Traveling to another community for abortion care can pose a barrier for anyone, and obstacles are likely compounded for many adolescents. For this population, even reaching a facility near where they live can be difficult if they do not have a driver’s license, access to a vehicle, or reliable public transportation [26]. In these situations, some adolescents may have no choice but to involve another person to help them obtain care, even if they would have preferred not to.

People obtaining abortion care in many states, including those where it remains legal post-*Dobbs*, have to navigate legal restrictions. These restrictions include mandated counseling and waiting periods (some of which may require two separate trips to the clinic), mandated ultrasound viewing, and gestational limits, among others [27]. In most states where abortion remains legal, adolescents under the age of 18 must also contend with parental involvement laws, which require that minors obtain permission from or notify one or both parents before they can obtain an abortion [28]. Research suggests that nearly two-thirds of adolescents under the age of 18 involve a parent in their abortion, even in the absence of these laws [29,30]. Still, minors who are not in contact with their parents, or who fear or live with mental or physical abuse, may be unwilling or unable to involve them. Judicial bypass, a legal procedure where minors can file to waive parental involvement in an abortion via court approval [28], is an option in states with parental involvement laws. While it is utilized by some minors, it can result in delays in accessing care or refusals by the court [31].

Any one of the above factors can delay access to care by days, weeks, or even months, and these barriers are often more pronounced, or compounded, for adolescents. These dynamics may explain the finding that adolescents are less likely than young adults to have a very early abortion (less than seven weeks gestation) [32] and why 18–19-year-olds are more likely than young adults to have second-trimester abortions [32,33]. Given the existing evidence of obstacles to abortion care faced by adolescents, this population deserves research into their experiences in order to reduce barriers to care.

## 2. Materials and Methods

### 2.1. Research Design and Data Collection

We used data from the Guttmacher Institute’s 2021–2022 Abortion Patient Survey (APS), a national sample of people accessing abortion care in the year prior to *Dobbs*, to examine the sociodemographic profiles of adolescent abortion patients as well as the financial and logistical aspects of care that they encountered. Our analyses focus on adolescents, or individuals under the age of 20 [4,5]. For comparison and to better understand if this population encounters different types of logistical and financial barriers, our final analytic sample (n = 6698) distinguishes between adolescents under age 20 (n = 633), young adults aged 20–24 (n = 2152), and adults aged 25 and older (top-coded to 45+, n = 3913). The data collection methodology has been described elsewhere [6,34] but we provide a brief summary below.

Using the Institute’s database of US abortion-providing sites, the sampling frame consisted of nonhospital facilities that provided 100 or more abortions per year, stratified by annual caseload. A specified number of facilities within each stratum were systematically sampled for recruitment. Between June 2021 and July 2022, staff at participating facilities recruited respondents by providing a flyer with a QR code or by obtaining an email or phone number from potential respondents at the time of their abortion visit.

Both recruitment mechanisms provided a link to an online Qualtrics survey that was available in English and in Spanish. The landing page contained a statement of informed consent and a continuation of the survey indicated that consent. The Guttmacher Institute’s Institutional Review Board granted a waiver of parental consent for minors, with the minor’s agreement to participate deemed sufficient. Respondents received a $20 electronic gift card upon completion. The final sample contains information from 6698 abortion patients at 56 facilities in 21 states. Participating facilities did not consistently track the number of patients who were not offered or declined survey participation, so we were unable to calculate a true response rate. Facilities in the sample reported providing 29,636 abortions during the survey fielding period, for an estimated participation rate of 23%.

Midway through the fielding period, we increased our efforts to recruit facilities (and respondents) in states likely to impose further abortion restrictions in response to the then-impending Supreme Court decision on *Dobbs*. As a result, individuals obtaining abortions in the Midwest and the South are overrepresented in our sample: 20% and 41%, respectively, compared to 16% and 36% of all US eligible abortions in 2020. We constructed a complex sampling weight to adjust for this overrepresentation as well as deviations from the original sampling design. These adjustments improve the alignment of the sample with the patient population in participating facilities; in the weighted data, people obtaining abortions in the Midwest and South represent 17% and 34% of the sample, respectively. However, we cannot claim that these adjustments are sufficient to make the data fully nationally representative. Our analyses focus on results using the weighted data; findings using the unweighted data are similar and are provided in the Appendix A.

### 2.2. Variables

Sociodemographic variables: The sociodemographic variables in the analyses include age, race/ethnicity, prior births, weeks pregnant, family income as a percentage of the federal poverty level (FPL), union status at the time of pregnancy, and type of health insurance coverage. Health insurance was categorized as Medicaid (including temporary Medicaid coverage for people who obtained this insurance specifically because they were pregnant even if they would not normally qualify), private insurance, government exchange (defined as insurance from HealthCare.gov, accessed on 16 February 2024 or a state-run health insurance marketplace or exchange), and uninsured. These characteristics have been associated with barriers to and differential experiences of abortion care in other studies [16,32]. All measures were based on respondents’ self-reports, and the measure of FPL was based on two items asking for family income and the number of family members the respondent lived with. We used hot-deck imputation to assign missing values to these variables. Most variables were missing information from less than 2% of respondents, except for poverty, where 11% of respondents did not provide information on family income or the number of family members. Notably, missingness on FPL was higher among adolescents in particular (16%) because they were less likely to report household income (12% vs. 6% of all other respondents). As a sensitivity analysis, we also generated findings using the unimputed FPL measure.

We included measures of sexual orientation and type of abortion (medication or procedural) as well, though these items were not imputed. Respondents from the full analytic sample who did not answer the items about sexual orientation and abortion type (n = 312 and n = 72, respectively) were excluded from analyses using each of these measures.

Measures for payment and logistics: We examined a number of payment-related measures (Table 1). These included: how the individual paid *for* the abortion, whether the respondent paid $0, the average amount paid for the abortion if greater than $0, actions taken to come up with money for care, ease or difficulty paying for the abortion, and if there were secondary costs for childcare, travel, and lodging. About one percent of the full sample did not answer the items about payment amount (n = 88) or whether it was easy/difficult to pay for the abortion (n = 76) and are excluded from analyses using these measures.

We also explored several measures related to travel and logistics, such as the type of transportation used to get to the abortion appointment and ease or difficulty traveling to the visit (Table 1); the latter excluded 97 individuals who either had their pills mailed to them (n = 9) or who did not respond (n = 88).

To understand how adolescents learn where to access abortion care, we examined responses to an item asking individuals why they decided to come to that specific facility. Respondents were provided with 14 options, and we examined responses reported by at least 10% of adolescents. Finally, respondents were asked if they would have preferred to have the abortion earlier and were provided with a list of 12 potential factors that could have led to delays in care; here, too, we limit analyses to reasons reported by at least 10% of adolescent respondents.

Because of the relatively small number of minors, or individuals under the age of 18, in our sample (n = 156, 2% of the sample), we do not examine them as a separate age group in most analyses. However, we provide a limited analysis comparing respondents aged 17 and younger to those aged 18–19 on several key sociodemographic characteristics (race/ethnicity, prior births, gestational age, FPL, and type of abortion), measures related to parental involvement laws (in the respondent’s state of residence or state of facility location), and other experiences that were reported by at least 50% of the adolescent sample.

### 2.3. Analysis

We first examined the sociodemographic profiles of adolescents, highlighting notable differences between this population and young adults and adult abortion patients. For each measure, we used simple logistic regression to test for significant differences between adolescents and the two other age groups. We adopted this same strategy for examining payment-, travel- and logistics-related measures. We also performed adjusted Wald tests to assess statistically significant differences in the mean amount paid for abortion care between adolescents and the two other age groups.

For the analysis comparing minors and adolescents aged 18–19, we calculated frequencies and corresponding 95% CIs to demonstrate the variability in the measures for these age groups, in addition to testing for significant differences using simple logistic regression.

We conducted all analyses using STATA SE 18 (College Station, TX, USA). We used the “svy” command to account for the complex sampling design and, where appropriate, used the “subpop” option. The study protocols were approved by the Guttmacher Institute’s Institutional Review Board.

## 3. Results

### 3.1. Demographic Profiles

Adolescents (<20 years) made up almost 10% of the sample, while young adults (20–24 years) composed about one-third (33%), and the majority, 58%, were adults aged 25 and older (Table 2). Among adolescents, the majority identified as non-white: 37% identified as Latinx, 23% as Black, and similar proportions identified as Asian or Other, 4% each. The majority of respondents were Black, Latinx, or Asian for all three age groups, but the proportion who identified as Black was lowest among adolescents and increased with age (23% vs. 32% aged 25+), while the proportion who identified as Latinx was highest among adolescents and decreased with age (37% vs. 26% aged 25+).

Nearly one-quarter of adolescents identified as something other than heterosexual, most commonly bisexual (17%), followed by 6% who identified as lesbian, pansexual or something else. Notably, as age increased, respondents were more likely to identify as heterosexual.

More than one in 10 adolescents, 12%, had had a prior birth. A substantial minority (29%) were married or cohabiting in the month they got pregnant, but the majority of adolescents were not living with a spouse or partner (71%). Some 12% of adolescent abortion patients reported that they were obtaining an abortion in the second trimester; notably, this decreased as age increased, and adolescents were significantly more likely than adults to be having an abortion after 12 weeks of gestation. There were no differences across age groups in obtaining an abortion at very early gestation (<6 weeks). Most adolescents obtained a medication abortion (58%), as did adults.

The majority of adolescent abortion patients, 55%, reported family incomes below the FPL, and nearly half had Medicaid health insurance coverage (48%). Sensitivity analyses examining the unimputed poverty measure show similar estimates as the imputed measure (Appendix A). Over a quarter of adolescents reported having no health insurance (26%), and nearly one in five had private coverage. As age increased, fewer respondents reported poverty-level incomes, and adults were less likely than adolescents to be uninsured.

### 3.2. Experiences Related to Payment

The majority of adolescents reported paying for their abortions out of pocket (54%); a substantial minority reported using any insurance (43%), most commonly Medicaid (31%) (Table 3). Slightly more than one in 10 adolescents reported relying on financial assistance (12%), such as an abortion fund or a facility discount. There were no differences in payment by age, with the exception that adolescents were slightly but significantly less likely than adults to report relying on financial assistance (12% vs. 16%). Over one-third of adolescents reported that it was very or somewhat difficult to pay for their abortion (34%).

On average, adolescents paid $499 for their abortion, and more than half (54%) reported having to do something to raise money, most commonly delaying other expenses (43%). One in 10 reported selling something (11%) to cover costs related to the abortion, and this proportion was significantly higher than for adults (7%).

### 3.3. Travel and Logistics

Two-thirds of adolescent respondents reported that someone they knew had driven them to the facility (66%), and the second most common mode of transport for this group was driving themselves (18%) (Table 4). While only 5% of adolescents reported taking public transportation, they were more likely than young adults to do so. They were also more likely than young adults and adults to have someone drive them to the facility. Still, two-thirds of adolescents reported that it was easy to reach the facility, and there were no differences by age group.

The top reason adolescents chose the health center they went to was because it was the closest (40%). About a quarter of adolescents chose the facility because it could see them soonest (26%) or because it offered medication abortion (25%). Eighteen percent of adolescents reported that they were at the facility where they obtained their abortion based on a friend or family member’s recommendation. The younger the age, the more likely respondents were to be at the facility based on this reason.

About two-thirds of respondents wanted their abortion sooner, and this proportion was consistent across age groups. Reasons for not obtaining their abortion earlier were wide-ranging. The majority of adolescents reported not knowing they were pregnant (57%), and 19% reported not knowing where to obtain the abortion as a reason for delay. One in 7 adolescents (16%) reported having to make travel arrangements, and 12% had to look into insurance. Type of insurance coverage was not associated with delays looking into insurance among adolescents, but this delay was reported more frequently among adolescents who relied on financial assistance (22%, *p* ≤ 0.05) (Appendix A). Compared to adults, adolescents were more likely to report not knowing they were pregnant (57% vs. 43%), not knowing where to obtain the abortion (19% vs. 11%), or that they were looking into insurance (12% vs. 5%).

### 3.4. Sociodemographic Characteristics and Experiences of Minors

Minors only accounted for 2.3% of the sample, but one-quarter of all adolescents. The majority of minors were 17 years old (55%), and 12% were 15 or younger (Table 5). Respondents under 18 were similar to adolescents aged 18–19 on many characteristics, with similar proportions identifying as Black or Latinx and having had their abortion in the first trimester. Not surprisingly, a smaller proportion of minors reported a prior birth as compared to 18–19-year-olds (7% vs. 14%).

More than half of minors received a procedural abortion, while the majority of older adolescents obtained a medication abortion. Although the confidence intervals overlapped, the associations were statistically significant. Additionally, compared to 18–19-year-olds, minors were more likely to have reported that somebody drove them to their appointment (80% vs. 62%).

The majority of minors, 53%, lived in a state with a parental involvement law. But when examining parental involvement laws for states where abortion care was received, a smaller proportion, 43%, obtained their abortion in a state with one, meaning that some minors who lived in a state with a parental involvement law travelled out of state for care. While the comparable figures for older adolescents were 53% and 51%, the difference between the two age groups for obtaining abortions in a state with a parental involvement law was not statistically significant.

## 4. Discussion

This study provides a profile of US adolescents as it pertains to several aspects of abortion care prior to *Dobbs*. In line with existing evidence to date, our findings suggest that adolescents were more vulnerable according to several domains [35,36]. The majority were Black or Latinx; in the United States, these populations are negatively impacted by institutionalized racism, or macrolevel systems that generate and reinforce inequities among racial and ethnic groups [37]. Nearly one-quarter of adolescents identified as non-heterosexual. The majority of adolescent abortion patients in our study were paying out of pocket for the abortion, at an average cost of $499. Many adolescents may not have their own income or may have part-time jobs. These factors may help explain why most had to do something to cover the cost of obtaining an abortion, including why they were more likely than older abortion patients to sell something. Most adolescent respondents reported family incomes below the FPL and were more likely than older abortion patients to do so; at the same time, as we discuss below, it is possible that this measure is less accurate for adolescents.

Several of our findings suggest that adolescents were more likely than adults to rely on other people when accessing abortion care. In particular, they were more likely to report that they went to the facility where they received care because they learned about it from a friend or family member, and two-thirds reported that someone they knew had driven them to their appointment. These circumstances do not necessarily represent obstacles, and for many adolescents, these contacts may serve as sources of support. Still, it is important to recognize that pathways to care differ for adolescents as compared to older individuals. For example, adolescents were more likely than older abortion patients to indicate that making travel arrangements led to delays in care. Many adolescents do not have access to a car to drive themselves and have to rely on another person’s availability to reach a facility. Some people who face criminal charges related to abortion in the US are turned over to law enforcement by someone who learns about the abortion [38]. Adolescents who have to rely on other people, especially when they do not want to, face potential legal risk for themselves and for the individuals providing them support due to restrictions such as parental involvement laws and the criminalization of self-managed abortion [38]. These factors all suggest that adolescents may need more logistical, financial, and legal support in the post-*Dobbs* landscape.

Two-thirds of adolescents in our study reported that they would have preferred to have had the abortion sooner, and they were similar to adults on this measure. However, they differed from older abortion patients in several of their reasons for delay in ways that suggest knowledge gaps. More than half of adolescents indicated one reason for the delay was not knowing they were pregnant, and they were more likely to report this than older abortion patients. Many adolescents have irregular periods, and a missed period may not be a cause for concern [23]. Alternately, some adolescents may not know or recognize the signs of pregnancy [22]. Adolescents were also more likely to report delays because they did not know where to obtain the abortion, which, in turn, may help explain why they were more likely to rely on friends and family members for recommendations. Our figures align with a nationwide study of adolescents and young adults that found that 17% reported not knowing where to seek information about abortion online [39]. Undoubtedly, the myriad of different abortion ban policies overlaid with the *Dobbs* decision have since made information about where individuals can seek abortion care more confusing, especially for adolescents.

Abortion stigma exists for people of all ages [40], and it is possible that, for this reason, some adolescents would have preferred not to have involved other people. Prior research has shown that adolescents have increased concerns about confidentiality when it comes to reproductive health care [18,19]. One mechanism for decreased confidentiality is the insurance explanation of benefits, as many adolescents obtain insurance coverage under their parents’ plans. More than one in 10 adolescents reported delays because they were looking into insurance coverage for abortion care. For example, some adolescents may not be aware of the type of, or if, they have insurance coverage. We found this delay was not related to the type of insurance coverage but was associated with relying on abortion funds, possibly indicating a desire to avoid insurance as a payment method. However, the number of respondents from which we were able to examine these patterns was quite small, and additional research is needed to fully understand them.

In our study, adolescents were more likely than adult abortion patients to be obtaining a second-trimester abortion, and this aligns with prior research [32,33]. This outcome is possibly due to any of the delaying factors discussed in the above paragraphs or the culmination of multiple factors. Adolescents with delayed recognition of pregnancy must also find a place to obtain an abortion and cover the costs, which are more expensive as gestation increases in a pregnancy [14,41]. Delays in either of the latter two factors could mean that an individual is more than 12 weeks pregnant by the time they access care. Recent research found that out-of-state travel for abortion care nearly doubled between 2020 and 2023 [42], which likely disproportionately impacts adolescents given their reliance on others for travel. Common delays to obtaining an abortion complicated by early gestation bans (6–12 week bans in particular) may make it especially difficult for adolescents in states with those bans to obtain care within that time frame. Removing gestational bans would help adolescents better access care, given their increased likelihood of obtaining a second-trimester abortion.

Adolescents under the age of 18 are a particularly vulnerable population because in many states, including those where abortion was legal in early 2024, minors must involve one or both parents in their abortion or undertake the onerous judicial bypass process [9]. Because of their small number, we were only able to examine a limited number of characteristics of this population. In our sample, most minors were 17, and they resembled adolescents aged 18–19 on many of the measures we were able to examine, including delayed recognition of pregnancy and payment. Still, minors were substantially more likely to have had someone drive them to the facility, and they were less likely to have a medication abortion. While prior research has found that adolescents are more likely to have medication abortions [34], the potential differences between minors and 18–19-year-olds are less studied. There is a need for more research differentiating within adolescents to better understand adolescents’ preferences and experiences.

Our study has several shortcomings. Although we created weights to better align our sample with the patient population of participating facilities, our data are not nationally representative. Further, information about adolescents on some measures may be less reliable. For example, many adolescents may not have knowledge of family incomes, leading to more missing information on this measure and less accurate answers when they are provided [43]. In addition, our sample includes too few minors to do a more comprehensive analysis of this population. Our study also does not capture people who wanted abortions but were unable to travel to a facility or those who were at a facility but declined to participate in the study.

## 5. Conclusions

The US has a history of stigmatizing pregnancy among adolescents. This study found that adolescents may encounter more risks to privacy or difficulties in accessing abortion care, and that 1 in 10 adolescents had a prior birth. It is important to recognize that abortion is not the preferred option for many adolescents. But it is also important to recognize that abortion is part of the full spectrum of sexual and reproductive health care for adolescents, and it should be treated as such under law and in standard medical practice. Adolescents have the right to have an abortion free from legal risk [44]. Efforts to protect and expand reproductive autonomy must include adolescents and respect their experiences accessing the full range of sexual health services, including abortion care.

## Figures and Tables

**Table 1 ijerph-21-00477-t001:** Wording for questionnaire items related to payment, travel and delays in accessing care, 2021–2022 Abortion Patient Survey.

Question Number	Item Wording	Response Categories
17	How will you or did you pay for this abortion? (Check all that apply)	Out of pocket but will be reimbursed by my health insurance company Private health insurance Medicaid or state health insurance Out of pocket (includes cash and credit cards) Financial assistance from an organization (includes abortion funds) Clinic price reduction or discount Other (please specify):
18	What is the total amount of money you paid or will pay to the health center for this abortion? (Include any amount that a partner, friends or family gave you and any insurance co-pay)	$ Amount: Not Applicable
20	Did you have to do any of the following to cover the cost of getting to the health center or paying for your abortion? (Check all that apply)	Sold something of value Delayed or put off other expenses Obtained a loan from a payday lender
19	How easy or difficult was it for you to cover the cost of this abortion?	Very easy Somewhat easy Neither easy nor difficult Somewhat difficult Very difficult
22	Did you lose wages from unpaid time off work in order to have this abortion? (If yes, please estimate lost wages)	Yes; $ Amount: No
21	To get this abortion, did you have to pay for any of the following? (If yes, please estimate amount you paid. If no, leave blank)	Transportation (round trip); $ Amount Paid: Lodging; $ Amount Paid: Childcare; $ Amount Paid:
23	How did you get to the health center where you obtained your abortion or abortion pills? (Check all that apply)	The pills were mailed to me I drove myself Someone I know drove me I took a taxi, Lyft, Uber, etc. I took a bus, train, or some other form of public transportation I took an airplane Other (please specify):
24	Thinking about the amount of time and travel, how easy or difficult was it to get to the health center where you got your abortion or abortion pills?	Very easy Somewhat easy Neither easy nor difficult Somewhat difficult Very difficult Not applicable, the pills were mailed to me
25	Did any of the below influence your decision to go to the particular health center where you got your abortion or abortion pills? (Check all that apply)	It was the most affordable It was the closest It takes my insurance It offers medication abortion (i.e., abortion pills, mifepristone) It offers surgical abortion (i.e., in-clinic abortion) It was recommended by another health care provider It was recommended by a friend, family member or someone I trust I have been here before It could see me the soonest It offers sedation It had positive online reviews They use telehealth services for abortion They provide curbside pick-up for medication abortion pill or provide pills by mail I am too far along in my pregnancy to go to other health centers
15	Would you have preferred to have this abortion earlier?	Yes No
16	Did any of the following prevent you from having this abortion earlier? (Check all that apply)	Not knowing I was pregnant Looking into or trying to get insurance coverage Not knowing where to get an abortion Getting time off work Making arrangements for childcare Making arrangements for transportation Coming up with the money to pay for the abortion Deciding whether to have an abortion This health center could not see me sooner I was told I wasn’t far enough along in the pregnancy I went to another health center for an abortion before I came to this one I had to delay or reschedule my appointment for reasons related to COVID-19

**Table 2 ijerph-21-00477-t002:** Sociodemographic characteristics of 2021–2022 Abortion Patient Survey respondents, by age group (weighted).

	All	Adolescents (<20)	Young Adults (20–24)	Adults (25+)	*p*-Values	
					Adol. vs. YA	Adol. vs. Adults
%	100	9.7	32.7	57.6		
n	6698	633	2152	3913		
	%	%	%	%		
Race/ethnicity						
Asian	4.2	4.0	3.7	4.5		
Black	29.4	23.4	27.5	31.5		***
White	30.1	30.9	28.1	31.0		
Other	6.6	4.4	6.4	7.1		
Latinx	29.8	37.3	34.3	25.9		***
Sexual orientation						
Heterosexual	83.9	77.6	80.6	86.8		***
Bisexual	12.1	16.8	14.9	9.8		***
Lesbian, pansexual, something else	4.0	5.6	4.5	3.5		*
Prior births						
0	45.3	87.9	62.0	28.6		
1 or more	54.7	12.1	38.0	71.4	***	***
Gestation						
<=5 weeks	18.3	17.9	17.6	18.7		
6–12 weeks	72.8	70.4	73.3	72.9		
13+ weeks	8.9	11.7	9.1	8.4		**
FPL poverty						
<100%	41.4	54.6	46.3	36.5	**	***
100–199%	30.3	27.3	30.2	30.9		
200+%	28.2	18.1	23.5	32.6	*	***
Insurance						
Medicaid	44.7	47.6	45.0	44.0		
Private	27.4	18.8	25.1	30.2	**	***
Exchange	6.3	7.3	5.5	6.5		
Uninsured	21.6	26.3	24.4	19.3		**
Union status						
Married	13.5	1.4	6.2	19.6	**	***
Cohabiting	35.7	27.4	38.8	35.4	**	**
Never/previously married	50.8	71.2	55.0	45.0	***	***
Abortion type						
Medication	55.6	57.5	58.2	53.8		
Procedural	44.4	42.5	41.8	46.2		

*p*-values from simple logistic regression: * < 0.05, ** < 0.01, *** < 0.001.

**Table 3 ijerph-21-00477-t003:** Payment-related characteristics for abortion care, by age group (weighted).

	All	Adolescents (<20)	Young Adults (20–24)	Adults (25+)	*p*-Values	
					Adol. vs. YA	Adol. vs. Adults
n	6698	633	2152	3913		
	%	%	%	%		
How paid ^						
Any insurance	41.8	43.3	41.2	41.9		
Private insurance	12.4	13.1	11.0	13.1		
Medicaid	29.9	31.1	31.0	29.0		
Out-of-pocket	52.8	53.7	53.9	52.1		
Financial assistance	14.7	12.3	13.6	15.6		*
Other payment method	2.4	1.7	1.9	2.7		
Paid $0	38.5	38.9	39.3	37.9		
Did any of the below to cover costs:	48.0	53.6	50.1	45.9		**
Delayed expenses	41.1	42.9	43.0	39.7		
Sold something	7.3	11.4	7.4	6.5	**	***
Payday loan	4.2	5.4	4.4	4.0		
Ease/difficulty of paying						
Very/somewhat easy	44.0	44.4	42.4	44.8		
Neither	21.9	21.1	23.3	21.2		
Very/somewhat difficult	34.1	34.4	34.3	34.0		
Had to pay for						
childcare	12.7	2.2	9.2	16.5	***	***
travel	31.9	28.7	29.8	33.6		
lodging	2.5	1.8	1.9	3.0		
Mean amount paid for abortion †	478.0	499.1	489.0	468.4		

^ Could indicate multiple responses; *p*-values from simple logistic regression: * < 0.05, ** < 0.01, *** < 0.001; † Adjusted Wald tests.

**Table 4 ijerph-21-00477-t004:** Travel and other logistical barriers to abortion care, by age group (weighted).

	All	Adolescents (<20)	Young Adults (20–24)	Adults (25+)	*p*-Values	
					Adol. vs. YA	Adol. vs. Adults
	%	%	%	%		
Transportation type						
Someone I know drove me	52.6	66.3	56.4	48.1	**	***
Drove myself	34.1	18.4	32.4	37.8	***	***
I took a taxi, Lyft, Uber, etc.	9.3	9.6	8.6	9.7		
I took a bus, train, or some other form of public transportation	3.4	5.1	2.8	3.5	***	
Travel ease						
Very/somewhat easy	69.1	66.1	68.7	69.8		
Neither easy nor difficult	15.9	17.1	15.7	15.7		
Somewhat/very difficult	15.1	16.9	15.6	14.5		
Reasons for going to this health center						
It was the closest	37.7	40.2	37.4	37.4		
It could see me soonest	26.2	26.4	28.2	25.0		
It offers medication abortion	22.8	24.8	23.6	22.1		
I have been here before	18.7	8.4	16.2	21.9	***	***
It offers surgical abortion	17.9	16.7	17.7	18.3		
It takes my insurance	14.6	14.8	14.7	14.4		
It was the most affordable	14.1	15.4	15.3	13.2		
It had positive online reviews	14.1	14.1	15.3	13.4		
It was recommended by a friend family member or someone I trust	12.5	18.1	14.1	10.6		**
Wanted abortion sooner	66.6	69.5	68.7	64.9		
Not know pregnant	46.4	57.2	49.4	42.7		***
Coming up with money	23.3	27.0	25.4	21.3		
HC couldn’t see me sooner	22.8	19.7	23.9	22.7		
Deciding to have abortion	21.7	22.6	23.1	20.7		
Getting time off	14.5	12.6	16.4	13.7		
Not know where to get abortion	13.3	19.2	14.8	11.4		**
Transportation arrangements	9.3	16.4	9.5	8.0	***	***
Looking into insurance	6.4	12.3	7.6	4.6	*	***

*p*-values from simple logistic regression: * < 0.05, ** < 0.01, *** < 0.001.

**Table 5 ijerph-21-00477-t005:** Characteristics of adolescents obtaining abortions in 2021–2022, by minors and 18–19 year olds (weighted).

	All <20	Minors (<18)	95% CI		18–19 Year Olds	95% CI		*p*-Value
								Minors vs. 18–19
N (unweighted)	633	156			477			
Age group								
14	1.8	7.6	3.9	14.3	na	na	na	na
15	0.9	4.0	1.6	9.6	na	na	na	na
16	7.7	33.0	24.7	42.5	na	na	na	na
17	12.9	55.3	45.3	64.9	na	na	na	na
Race/ethnicity								
Black	23.4	24.8	16.5	35.4	23.0	18.0	28.9	
White	30.8	31.8	21.1	44.9	30.5	24.6	37.1	
Other	8.4	5.2	2.4	11.2	9.4	5.9	14.7	
Latinx	37.4	38.2	27.3	50.4	37.1	30.7	44.0	
Prior births								
0	87.9	93.1	87.3	96.3	86.4	81.4	90.2	*
1 or more	12.1	6.9	3.7	12.7	13.6	9.8	18.6	
Gestation								
≤12 weeks	88.3	87.1	82.3	90.8	88.6	83.6	92.3	
13+	11.7	12.9	9.2	17.7	11.4	7.7	16.4	
FPL poverty								
<100%	54.6	64.1	53.2	73.7	51.7	45.2	58.1	
100–199%	27.3	24.7	18.1	32.7	28.1	24.3	32.3	
200+%	18.1	11.2	6.4	19.0	20.2	15.0	26.6	
Abortion type								
Medication	57.5	46.7	34.0	59.8	60.7	49.4	71.0	*
Procedural	42.5	53.3	40.2	66.0	39.3	29.0	50.6	
Paid OOP	53.7	52.3	39.3	65.0	54.1	43.4	64.5	
Did any of the following to cover costs: delayed expenses, sold something, payday loan	53.6	56.4	45.5	66.8	52.8	47.1	58.4	
Transportation type								
Someone I know drove me	66.3	80.0	69.3	87.6	62.1	54.1	69.5	**
Travel ease								
very/somewhat easy	66.1	63.1	56.2	69.5	67.0	60.8	72.6	
Wanted abortion sooner	69.5	69.7	58.7	78.8	69.5	63.9	74.5	
Not know pregnant	57.2	53.9	44.8	62.6	58.2	50.9	65.1	
Lives in state with parental involvement law	53.3	53.2	37.8	68.1	53.3	39.6	66.6	
Clinic in state with parental involvement law	49.1	42.5	27.8	58.7	51.1	37.4	64.6	

*p*-values from simple logistic regression: * < 0.05, ** < 0.01, na is not applicable.

## Data Availability

The data presented in this study are available on request from the corresponding author. The data are not publicly available due to confidentiality reasons.

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
