# Peer review of "Characteristics and Circumstances of Adolescents Obtaining Abortions in the United States"

_ijerph, 2024, doi:10.3390/ijerph21040477_

Round 1
Reviewer 1 Report
Comments and Suggestions for Authors
Thank you for the opportunity to review this paper, which is very well-written and positioned to make a noteworthy contribution to the literature on abortion seeking among a diverse sample of US adolescents. Below, I have outlined a few minor points, mainly about improving transparency and clarity about the study sample, that could be addressed to further strengthen this manuscript for publication in IJERPH.
1. Abstract: It is unclear in the abstract whether you are conducting analyses using only the adolescent sample or whether you are also comparing adolescents and adults in the larger overall sample. Based solely on the abstract, I thought it to be the former, but when you state “Adolescents differed from adults in their reasons for delays in accessing care” it seems that you have run analysis on this point across age groups. Could you add a touch of detail to the abstract to make it clear whether this comparison is part of your current analysis or whether you are comparing your current sample (adolescents) to existing research on adults?
2. Page 2, Lines 70-74: There are likely other factors coming into play here, too. I’m thinking of how adolescents may be less likely to recognize other early signs of pregnancy due to inadequate sex education, social networks with less experience with pregnancy, and a shorter personal history of sex and reproduction.
3. Page 2, Line 78: Edit to “aged 15-44 years” for clarity.
4. Page 3, Lines 142-144: Similar to my point above about the abstract, I found this statement a bit confusing. I understand the importance of your comparative analysis, but I think your exact methods could be communicated with a touch more precision. First, please provide the sample sizes in this paragraph. Also, please read through the methods section for opportunities to improve the consistency of this messaging. Is your sample 633 adolescents or all 6698 abortion seekers? What is the age breakdown of the remaining 6065 people? What is the upper age limit for the 25 and older group? I also question whether the variables section is the right place for this important information about your study sample which seems to otherwise be described in the preceding section.
5. Page 3: Related to my point above, the size of your analytic sample is unclear. Is the 633 adolescents you mention on Line 109 your analytic sample or is it much smaller once you remove all the missing data described in the variables section? Does the missingness you describe throughout page 4 come from the full 6698 sample? This has important implications. Initially, as I was reading, I thought it was coming out of the sample of 633 adolescents, making some of these numbers alarmingly high (e.g., 312 people removed due to missing sexual orientation information). Some of this information is available once I get to the Tables, but it should be clear in the methods section as well.
Reviewer 2 Report
Comments and Suggestions for Authors
1- according to which definition, adolescent is referred to people under 20 years old in this study? The title of the manuscript and the introduction needs to be rewritten.
2- Why do you include young adults, aged 20-24, and adults, aged 25 and older? Are they a control group? Is your study a case-control?
3- How was the observance of ethical principles?
Reviewer 3 Report
Comments and Suggestions for Authors
In this manuscript, the authors use a survey of 6,698 US abortion patients, conducted between 2021 and 2022, to compare demographic characteristics of, along with financial and practical support used by, adolescents (<20 years of age), young adults and adults ≥25 years old. The authors found significant differences between adolescents and patients aged 20 or older on gestational duration at abortion, income and insurance coverage, some indicators of financial and travel logistics for abortion care, and reasons for abortion delay. There were few differences between minors and adolescents 18-19 years old, other than having a prior birth and abortion type.
This is an informative profile of younger abortion patients, who face different challenges obtaining abortion care, and adds to the literature on this topic. The separate analyses for minors and younger adolescents 18-19 years of age is useful. Given that the journal’s readership may be less familiar with the US abortion context, additional details about some of the policy context are needed throughout. The following changes would strengthen the manuscript.
Abstract
1. The comparisons between adolescents and young adults/adult patients could be clearer in the Results so that there is more support for the concluding statements about adolescents being more vulnerable and having ‘unique barriers.’
Introduction
2. The first paragraph of this section could be reworked to make the flow of ideas easier for readers to follow. It may be helpful to move the statement about the needs of adolescents after the section about the potential impacts of the Dobbs decision.
3. More updated sources, including the American Academy of Pediatrics and US Office of Adolescent Health, use a wider age range in their definition of adolescence. To align the authors’ age groups with these definitions, it may be useful to refer to abortion patients <20 years of age as younger adolescents.
4. Because not all readers of this journal will be familiar with the Hyde Amendment and the limitations it imposes, it would be helpful if the authors could provide a brief description.
Methods
5. Pg 4, line 149. By temporary Medicaid, do the authors mean Medicaid that someone had during pregnancy?
Results
6. Pg 5, line214. BIPOC has not been defined and many readers outside the US may not be familiar with this acronym. Because the authors only use this a few times in the text, it may not be worth including and instead they can refer specifically to the racial identities of the groups of interest.
7. Pg 12, line 293. There appears to be a stray letter in this sentence.
8. Tables 2 and 5. Change the row heading “Procedure type” to “Abortion type” (or Abortion method”).
Discussion
9. Can the authors provide more social context about why racial and ethnic identity may make adolescents more vulnerable than young/older adults?
1. Pg 13, lines 314-321. The link between transportation needs to legal risks is not as clear as it could be. Also, it seems important to acknowledge why adolescents may be more likely to have someone drive them and why this creates a delay (i.e., that they don’t have their own car and have to rely on someone’s else’s schedule to get to a facility for care).
1. Pg 14, lines 342-345. The delays associated with insurance may also be related to the fact that adolescents do not even know what type of insurance they have, and not only a desire to avoid using their insurance for their abortion. This also should be acknowledged.
1. Pg 15, lines 387-392. These statements, although important, start to take the conclusions in a different direction. If the authors wish to retain these ideas, the statements could be rephased so that the flow of ideas is smoother. Perhaps something along the lines of “Although abortion is not the preferred or best option for all adolescents, they do have a right to abortion care that is free from legal risk...”
Comments on the Quality of English LanguageNone
